# Characterizing the Stormwater Runoff Quality and Evaluating the Performance of Curbside Infiltration Systems to Improve Stormwater Quality of an Urban Catchment

Hussain Shahzad [1],*, Baden Myers [1], Guna Hewa [1], Tim Johnson [1], John Boland [1] and Hassan Mujtaba [2]

1   Unisa STEM, Mawson Lakes Campus, University of South Australia, Adelaide, SA 5095, Australia; Baden.myers@unisa.edu.au (B.M.); Guna.hewa@unisa.edu.au (G.H.); Tim.Johnson@unisa.edu.au (T.J.); John.Boland@unisa.edu.au (J.B.)
2   Department of Civil Engineering, University of Engineering & Technology, Lahore 54890, Pakistan; hassanmujtaba@uet.edu.pk
*   Correspondence: hussain_mustafa.shahzad@mymail.unisa.edu.au

**Abstract:** The conveyance of stormwater has become a major concern for urban planners, considering its harmful effects for receiving water bodies, potentially disturbing their ecosystem. Therefore, it is important to characterize the quality of catchment outflows. This information can assist in planning for appropriate mitigation measures to reduce stormwater runoff discharge from the catchment. To achieve this aim, the article reports the field data from a typical urban catchment in Australia. The pollutant concentration from laboratory testing is then compared against national and international reported values. In addition, a stochastic catchment model was prepared using MUSIC. The study in particular reported on the techniques to model distributed curbside leaky wells with appropriate level of aggregation. The model informed regarding the efficacy of distributed curbside leaky well systems to improve the stormwater quality. The results indicated that catchment generated pollutant load, which is typical of Australian residential catchments. The use of distributed storages only marginally improves the quality of catchment outflows. It is because ability of distributed leaky wells depended on the intercepted runoff volume which is dependent on the hydrological storage volume of each device. Therefore, limited storage volume of current systems resulted in higher contributing area to storage ratio. This manifested in marginal intercepted volume, thereby only minimum reduction in pollutant transport from the catchment to outlet. Considering strong correlation between contributing impervious area and runoff pollutant generation, the study raised the concern that in lieu of following the policy of infill development, there can be potential increase in pollutant concentration in runoff outflows from Australian residential catchments. It is recommended to monitor stormwater quality from more residential catchments in their present conditions. This will assist in informed decision-making regarding adopting mitigations measures before considering developments.

**Keywords:** stormwater quality; field investigations; stormwater systems; catchment model

## 1. Introduction

Rapid urbanization has resulted in increased stormwater runoff flowrates and volumes. The management of this increased urban stormwater runoff is a growing concern for catchment managers, considering its harmful effects for receiving water bodies. In the literature [1–3], there is a consensus that stormwater discharge from residential catchments is a major pollutant source to receiving surface bodies. Urban stormwater carries urban pollutants with it to the catchment outlet. The urban pollutants in general consist of sediments, oxygen-demanding substances, heavy metals, organics, bacteria, viruses, nutrients, litter, and natural organic matter [4]. Total nitrates (TN) Total Phosphorous (TP), and TSS (Total suspended solids or sediment) are pollutants of particular concern to many proponents of policy for urban stormwater. Excessive nutrients in the form of TP and TN loading in receiving waters can lead to cultural eutrophication and algal proliferation [5]. TSS and nutrients

can damage the ecology of receiving waters—for example, coastal reefs, which support a variety of fish, molluscs, and seastars [6]. TSS can also reduce the hydraulic efficiency of receiving waters as sediment continuously accumulates. In Australia, urban stormwater is conveyed by a separate drainage system. This means that stormwater is conveyed to receiving water bodies and typically discharges from catchments without treatment.

Urban planners and policymakers need economical and feasible solutions to this problem. Reducing discharge of stormwater runoff and pollutants from catchments before discharge to receiving waters is one way of managing the harmful effects of urban stormwater runoff [7]. For this, different studies, for example, Todeschini et al. [8], have attempted to characterize the performance of different management options to improve the quality of runoff discharges from the catchment to protect the quality of receiving water bodies. Green spaces provide natural infiltration losses and depression storages to hold rainfall runoff. However, with rapid urbanization, there are limited green spaces available in cities for limiting outflows from catchments [9]. Urban planners worldwide are adopting a variety of approaches to mitigate problems with urban stormwater. These approaches are applied all over the world with different names. For example, in UK Sustainable Urban Drainage Systems (SUDS), in the USA Best Management Practices (BMP), and in China "sponge cities" [10], while the current study, based in Australia, adopts the term Water Sensitive Urban Design (WSUD). These approaches to urban design have shown tremendous potential, and the literature has reported successful case studies for managing urban stormwater [7]. Infiltration systems are one of the constructed forms of WSUD measure for reducing runoff and pollutant volume. Infiltration systems cover broad array of devices ranging from source control devices (e.g., leaky wells) to more catchment scale devices like bioretention systems [11].

One popular method of retrofitting established catchments with infiltration systems is the use of distributed infiltration systems, for example, leaky wells [12], soakaways [13], and bioretention systems [14]. In this strategy, infiltration systems are distributed over the catchment to intercept contributing impervious area. These devices, due to their small footprint, can easily be retrofitted in an existing catchment and have a demonstrated potential to reduce runoff volumes from urban catchments [14]. Transport of stormwater pollutants is the function of runoff outflows from the catchment [15]. By reducing runoff volume, distributed infiltration systems can also contribute to reducing the transport of associated pollutant loads proceeding downstream thus providing solution to protect receiving water bodies. Literature has reported their effectiveness in reducing stormwater runoff, e.g., Locatelli, Mark, Mikkelsen, Arnbjerg-Nielsen, Deletic, Roldin, and Binning [11]. However limited research is available, which has drawn conclusions from the monitored field data, regarding the efficacy of these devices to manage urban runoff and water quality at the catchment scale [16].

This lack of knowledge means urban planners can be hesitant to prescribe infiltration measures to reduce runoff and meet water quality targets [16]. As an alternative to field studies, hydrological modeling offers cost-effective way of establishing the usefulness of distributed leaky well systems [17]. In this domain, use of simple stochastic hydrological models like the Model for Urban Stormwater Improvement Conceptualization (MUSIC) [18], a commercial software developed for modeling WSUD devices and Australian catchments, can prove useful to quantify catchment outflows and the performance of constructed WSUD devices which may be implemented. In Australia, design practitioners have used MUSIC to support the implementation of WSUD systems in the catchments. Default input values for pollutant concentrations according to land use in MUSIC are based on the findings of an extensive review of stormwater quality in urban catchments performed by Duncan [19], and more localized parameters are also available [20]. Another issue specific to modeling of distributed system is that the modeling of a large number of distributed systems individually can become a very tiresome and complex process. In MUSIC, large numbers of WSUD systems, for example, rainwater tanks [21], can be aggregated to form a lumped model, where multiple systems are represented as a single larger

device. However, aggregation causes issues; becasue when a large quantity of devices is aggregated in MUSIC, the model has been reported to falsely exaggerate their performance levels. Elliott, Trowsdale, and Wadhwa [21] has discussed the details of problems associated with the aggregation of infiltration devices. While the modeling of each individual system is tiresome, the conclusion drawn from the research [21] is that aggregation of devices should be within the model limitations, so as not to affect the performance of distributed infiltration devices.

*Research Objectives*

The focus of this study is to characterize the stormwater quality of a residential urban catchment in South Australia, of which little has been reported with respect to individual land uses. The information will serve the purpose of informing policy makers regarding the estimated urban pollutants, a typical catchment discharge during a storm. Further the study will place the case study catchment in the context of the national and international averages for residential stormwater pollution. This will explain that if the catchment is indeed a representative of typical Australian catchment in terms of pollutant discharges. The study planned to achieve this through comparison of stormwater quality from study catchment against reported national and international averages of pollutant concentrations in stormwater runoff discharges from urban catchments.

An additional goal of the study is to demonstrate the effectiveness of MUSIC to simulate runoff from the catchment by carrying out statistical evaluation. This evaluation has not been widely reported in the literature. This information will benefit potential users of MUSIC to consider automated calibration before applying MUSIC in practice. Further, results could inform the readers regarding representing the smaller curbside leaky well systems in MUSIC. This will also include discussion regarding the appropriate level of aggregation for modeling curbside leaky well systems, without falsely simulating their performance to reduce the stormwater pollutant loading. In doing so, the study established the capability of MUSIC to characterize the stormwater quality from urban catchment. Based on the MUSIC simulations, the study also aims to inform reader regarding the efficiency of curbside leaky well systems to reduce the annual pollutant load from the catchment outflows. The results of the study will inform policymakers regarding the potential of distributed infiltration measures to reduce pollutant loads.

The results of this study will provide important information to planners regarding the current state of catchment stormwater quality, how it is represented in common decision-making tools and how it may be improved. The results for the case study area considered are intended to contribute to ongoing development of effective policy regarding urban stormwater more broadly and improve the design of distributed stormwater systems.

## 2. Materials and Methods

The methods described here cover the procedure to extract data and develop a MUSIC model of the case study catchment (Figure 1). The methods will describe field data and laboratory testing programs and the methods used to derive a calibrated and verified MUSIC model of a case study catchment. The methodology will also describe model calibration method. In addition, the modeling section will provide insight to modeling of distributed curbside infiltration systems. Figure 1 provides with an overview of the methodology adopted to achieve the objectives of this study.

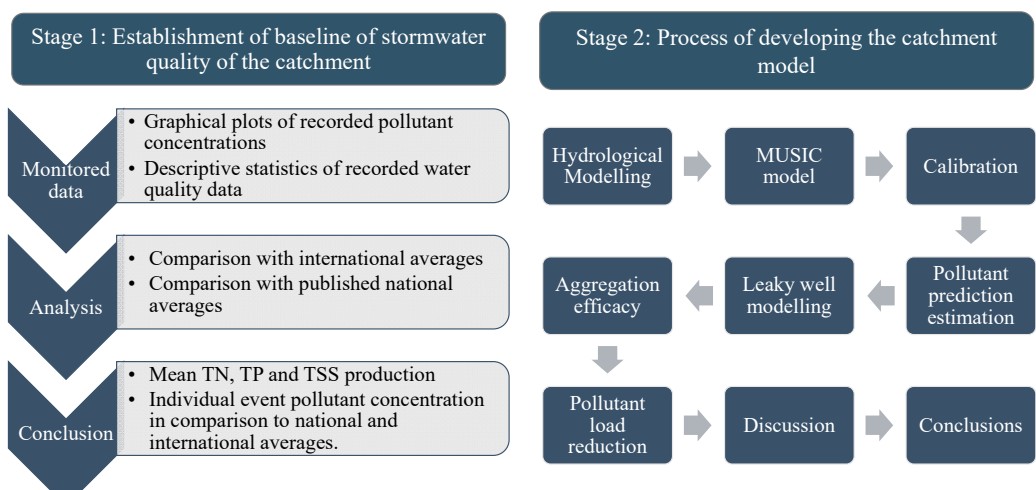

**Figure 1.** An overview of the methodology adopted to carry out the study.

### 2.1. Catchment and Curbside Leaky Well Description

Field investigations and monitoring were conducted in a 17.45 ha urbanized sub-catchment comprising of 108 residential allotments and road infrastructure in Hawthorn, an inner suburb of Adelaide, the capital city of South Australia (Figure 2). The catchment consisted of established homes on well vegetated allotments. The average allotment size was 700 m². The local climate is semi-arid, with most of the average annual rainfall of 541 mm falling between late autumn and early spring (May to September). Approximately 80% of precipitation typically falls at intensities of less than 4 mm/h. Average annual potential evapotranspiration was 1500 mm [22] with hot, dry summers (December to February). The catchment terrain grades evenly in a westerly direction with a gradient of 0.5%. The catchment had a separate sewer and stormwater drainage system, as is usual in Australia. The catchment followed the standard curb and gutter stormwater collection practices. Each house in the catchment discharged to the street gutter and stormwater was carried along the street leading to a catch basin at the end of the street to the underground stormwater pipe network. Stormwater drainage pipes ranged from 225 mm to 450 mm in diameter.

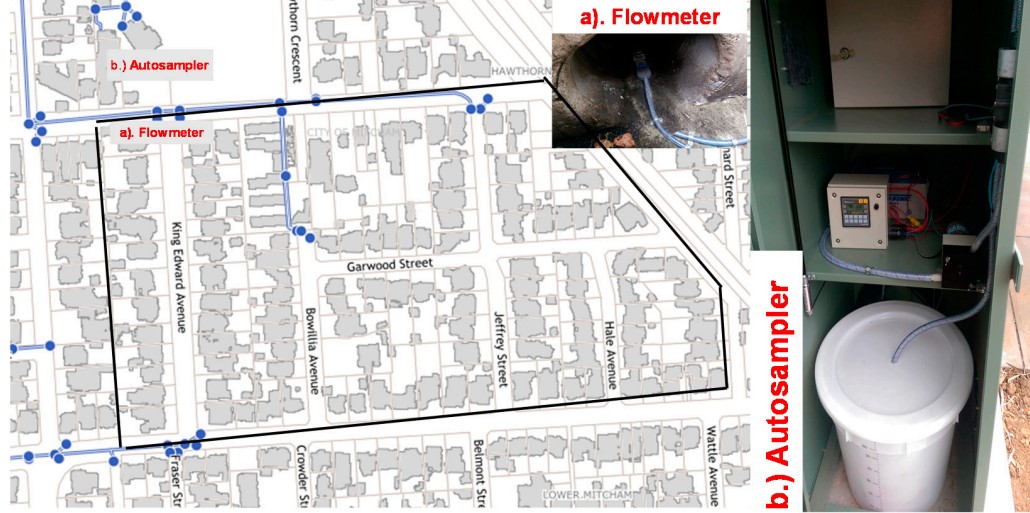

**Figure 2.** The case study catchment with drainage network is shown. The extent of the catchment boundary is highlighted and inset. (**a**) The flowmeter installed at the end of catchment outlet pipe and (**b**) the automated water sampler enclosed in a cabinet.

Adelaide has cooler temperate waters in its coastlines, which offer a conducive environment for the growth of macroalgae. In fact, Adelaide coastline supports 30–40 percent of total macroalgae species exists in the world [6]. Nutrients—the source of nitrogen, can cause algal blooms and epiphyte growth on seagrass, leading to loss of seagrass. Seagrass meadows are of fundamental importance to the ecosystem in Gulf St Vincent [6]. They bind the sediments and provide nurseries and safe habitat for marine organisms. Loss of seagrass will also result in erosion of beach soil, resulting in degradation of coastline. MacDonald, Ardeshiri, Rose, Russell, and Connell [6] reported the loss of one-third of original sea grass in over 80 years due to continuous expansion of urban areas. Similarly, discharges of high levels of suspended solids into the coastal waters increase turbidity levels contributing to poor recreational water quality and may result in beach closures. It is understood that stormwater nutrients, turbidity and sediments may have been a contributing factor to seagrass die-off. Stormwater is the major contributor (67%) to sediment load discharged into the coastal environment. Due to these mentioned issues associated with stormwater, it is acknowledged as having an adverse impact on receiving waters. However, it is not considered financially viable to discard the standard practice of discharging stormwater without treatment to receiving water bodies. Based on the problems associated with urban stormwater quality, this study has attempted to describe the stormwater quality in typical Australian residential catchment in Hawthorn, an inner suburb of Adelaide, South Australia.

*2.2. Monitoring Equipment*

In December 2015, the catchment was equipped with monitoring instruments to measure the quantity and quality of stormwater discharge. A tipping bucket rain gauge (TB3, Hyquest Solutions, Warwick Farm, NSW, Australia) was used to collect rainfall data in the case study catchment at a one-minute resolution in increments of 0.2 mm. An area-velocity flow meter (Starflow Ultrasonic Doppler Instrument, Model 6526, Unidata Pty Ltd., O'Connor, WA, Australia) was installed in the 450 mm diameter concrete drainpipe at the catchment outlet, measuring stormwater runoff flow rate and volume from the 17.45 ha catchment. Depth measurement accuracy was $\pm 1$ mm; flow velocity accuracy was $\pm 1$ mm/s. Rain and flow data collection started on 21 December 2015 and was ongoing at the time of writing.

Water quality of runoff form the catchment was measured using an autosampler (Water Data Services flow proportional composite sampler, Adelaide, Australia) including a peristaltic pump connected to a sample hose and composite sample tub of 60 L capacity, for the collection of samples for water quality analysis. The sampler was connected to a controlled programmed to collect 500 mL samples for every 5000 L of discharge in the drainpipe. The typical depth for sample extractions was set to 50 (mm) to 150 (mm). At the conclusion of a storm, the tub contains a composite water quality sample proportional to the flow. Analysis of the composite sample provides a flow weighted mean concentration of each pollutant and combined with runoff volume, enables the determination of event pollutant load.

*2.3. Curbside Leaky Well Installation*

The installation of curbside leaky well systems was part of the City of Mitcham initiative to reduce runoff volume from the catchment, to protect downstream receiving water bodies. Curbside leaky wells were installed in street verges—local government green space between the road and footpath. The depth of leaky wells varied from 820 to 1400 mm with width of 460 mm occupying total surface area of 120 mm$^2$. In total 181 leaky wells were installed, amounting to 10 systems per hectare of catchment. The TREENET curbside inlet flow capture device consists of a slotted face plate and a PVC pipe fitting that are cast into the concrete of the curb and gutter (Figure 3). The plate has been designed to restrict the inflow of leaves and other litter through to the leaky well. The design of the inlet included a shallow basin in the gutter which created a small pool from which the

water flowed into the inlet. Reduced flow velocity due to greater cross-sectional area in the gutter established an eddy in the pool which deposited larger sediments away from the capture slot. In this way, clogging of the inlet may be reduced and routine mechanized street sweeping used to remove deposited sediment [23].

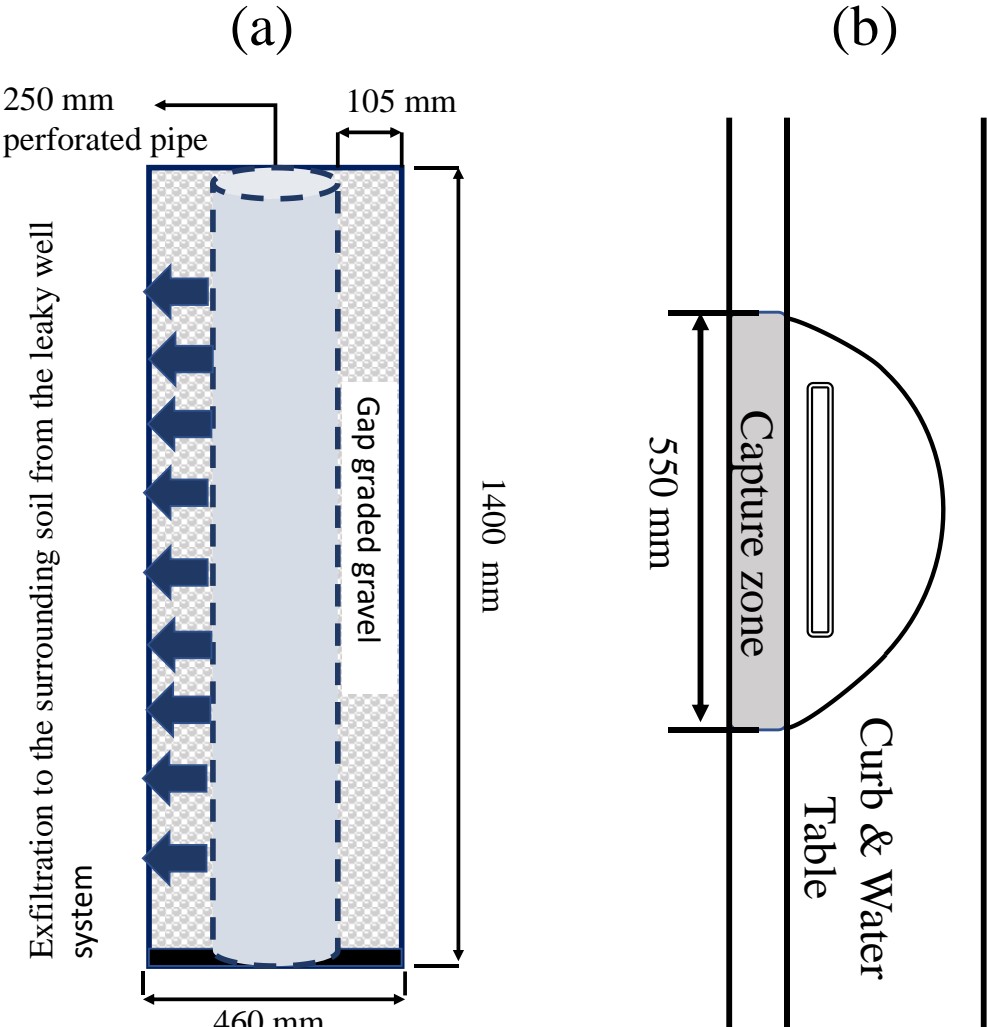

**Figure 3.** The layout of curbside leaky well systems. (**a**) The cross section of installed leaky well systems. The dimensions of the leaky wells are shown along with description of materials used (**b**). The face plate of R750 Treenet inlet is shown with capture zone.

*2.4. Description of Data Collection*

Rainfall and runoff were recorded in one-minute resolution. However, for this study, rainfall and runoff data were manipulated to produce data in six-minute resolution. This was required to construct a model of the catchment using MUSIC version 6.1. Data up to December 2016 illustrated catchment behavior without curbside leaky wells (preinstallation). In this time, six water quality samples were obtained. One of these samples was not included in the analysis due to construction activities in the catchment during the time of sampling causing an unusually high level of sediment.

*2.5. Catchment Model Development*

In this study, we have developed a model of the case study catchment using MUSIC Version 6.1 [18] model to meet our study objectives. The details of MUSIC are provided in program documentation [18]. Here, we briefly describe its main feature to provide context to the model building for this current catchment. In MUSIC, the modeler can represent

the catchment using number of source nodes. However, the ability of MUSIC to represent the drainage network is not as developed as other models more focused on hydraulic conveyance, e.g., EPA SWMM. Source nodes, used to represent a subcatchment, have homogenous soil properties, impervious cover, and pollutants generation. Each source node also has groundwater reservoir option as well.

The runoff predictions are based on user defined rainfall and evapotranspiration data. Each sub catchment, when its infiltration capacity is exceeded, discharges to the outlet through links.

### 2.5.1. Initial Parameter Section

Initially, we constructed a model of the case study catchment with eight subcatchments. The decision to represent the catchment with eight nodes (lumping lots together) as opposed to 108 nodes representing each home was made to reduce model runtime (35 min to 3 min) and complexity. There are three main parameters in MUSIC to characterize a subcatchment: total area, imperviousness, and perviousness. We aggregated total area as the sum of lot area in each node, which were part of the aggregated subcatchment. For imperviousness and perviousness, we took the weighted average for individual house in the aggregated catchment. The values for these parameters were directly obtained from investigating aerial photography in a graphical information system. Next, we estimated the rainfall threshold parameter based on calibration, seeking to ensure the commencement of runoff flow correctly during the storm. The rainfall threshold parameter was found to be influential in describing the shape and timing of the hydrograph. The model parameters adopted to construct the case study catchment relied heavily on SA Guidelines for MUSIC modeling [20], specific for Adelaide region. The ground water contribution in the model were ignored as only surface runoff flow data was available. Table 1 contains the final set of parameters adopted to represent the runoff generation capacity of the catchment. The rainfall–runoff parameters, in particular, soil parameter values, we selected for calibrations was based on suggestion of SA Guidelines for MUSIC modeling [20], to calibrate these values in the presence of available flow data. The model was calibrated to preinstallation runoff flow data. It is because that focus of the study is to characterize the stormwater quality of the catchment without treatment measures. This will allow the review of total pollutant generation without the treatment measures, which can then be compared with future development scenarios.

### 2.5.2. Automated Calibration

We used Parameter estimation software (PEST) version 17 [24] to calibrate runoff generating parameters of the catchment. PEST is an automated method of estimating the parameters, with focus to reduce the objective function based on finding the local optima. PEST uses a gradient-based linear approach in finding local optima by using the Gauss–Marquardt–Levenberg method [24]. We established the MUSIC/PEST interaction by running the basic MUSIC input file (msf) from the command line. In addition, MUSIC configuration file (mcf) was also prepared and provided in the executable command in PEST. The configuration file defines the data, to be extracted from MUSIC outputs, and also the location where these outputs will be written. The outputs from MUSIC include time series, mean annual loads, node water balance, and statistics, which can be extracted from any node and then written to a text file. The PEST algorithm changes the initial, user-defined calibration parameters in a MUSIC input file over successive model runs to optimize the fitness of the model to observed data. In doing so, it created a new MUSIC input file with the calibrated set of parameter values. In this calibration we adopted the Taylor series expansion to linearize the process. In this method, the partial derivatives from each model run are evaluated with respect to every parameter change after each iteration. The outputs from this iteration are the current optimal set of parameters. PEST then compares the parameter set to that of optimal set as obtained from previous iterations. If three iterations passed without significantly lowering the objective function PEST, then

terminates the estimation process. PEST offered options for users to provide weight to the individual events. Doing this, the objective function is influenced by these weighted events. Therefore, PEST focuses on getting those values right to lower the objective function. Objective function in PEST is the difference in model prediction and weighted observed runoff flow at any point in time. Equation (1) provides the mathematical form of the objective functions as used in PEST.

$$\phi = \sum_{t=1}^{m} (Ot.Rt)^2 \qquad (1)$$

where $\phi$ = objective function, $Ot$ = Weighted observed runoff flow at time $t$, $Rt$ = residuals from weighted observed, and simulated runoff flows in time $t$.

**Table 1.** The list of parameters adopted to represent the catchment. The reference to relevant section of SA modeling guidelines for MUSIC [20] is also provided.

| Parameter | Value | Method of Estimation | SA Guidelines for MUSIC Modelling [20] |
|---|---|---|---|
| Sub catchments | 8—Urban nodes | Lumped Approach | Section 4.1 |
| Area (ha) | 0.8 to 4.04 | GIS map available | |
| Imperviousness | 28~43% | GIS map. The imperviousness was based on the subtraction of indirectly connected impervious area total impervious, for conservative estimate. | Section 4.2.2, recommends estimating for impervious fractions based on available plans |
| Rainfall threshold (mm/day) | 4.0~5.0 | Calibration using PEST, Values varies from 1 mm/day to maximum of 5 mm/day as recommended in MUSIC manual | No specific information is available to select this parameter |
| Soil storage capacity (mm) | 102 | Used in calibration by providing range of 88 mm to 108 mm, based on recommended values for light clays in SA Guidelines for Music modelling | Section 4.3 |
| Field capacity (mm) | 69 | Used in calibration by providing range of 63 mm to 83 mm, based on recommended values for light clays | Section 4.3 |
| Infiltration capacity coefficient—a (mm/day) | 145 | Used in calibration by providing range of 125 mm to 145 mm, based on recommended values for light clays | Section 4.3 |
| Infiltration capacity coefficient—b (mm/day) | 0.5 | Used in calibration by providing range of 0.5 mm to 4 mm, based on minimum and maximum values for different soil groups | Section 4.3 |

PEST also estimate the partial derivatives of model outputs at each iteration using central finite differences. PEST then estimates the sensitiveness of each parameter as by product based on these derivatives.

In this calibration, we assigned more weight to the representative flow events and assigned "0" weight to runoff flow events which arise as result of extreme events and to no-flow events. It is important to simulate every day storm events correctly as they are the source of the majority of the runoff producing storms and therefore have cumulative effect on total runoff of the simulated series. Due to impact of runoff volume influence in simulating the pollutant, it was deemed important to model the catchment with focus to predict runoff volume correctly. We adopted the option of log-transformation of parameter values during the inversion process. It is because adopting logarithmic transformation of model parameters, increase the ability of PEST to hold its linearity approximation in case of nonlinear problems [24].

At the end of inversion process, PEST provides the optimized set of parameter values rather than its logarithmic values. PEST also provides as an output 95% confidence limits of optimized parameter values. PEST also provides as an output 95% confidence limits

of optimized parameter values [24]. Only few prior studies, for example, Dotto et al. [25] have reported the uses of PEST for calibrating MUSIC. Five parameters were calibrated to meet the runoff flow series in PEST.

### 2.6. Model Evaluation

We evaluated the catchment model calibration and validation following similar techniques as recommended by the ASCE Task Committee [26], i.e., first by visual inspection of graphical outputs, then by comparing statistical indices for the continuous runoff series, and finally by computing goodness-of-fit statistics for individual events extracted from the continuous simulated flow series. The ASCE task committee [26] recommended that to assess the performance of a continuous simulation model, total percentage error in volume (PEV), Nash Sutcliffe Efficiency (NSE), and the coefficient of gain from daily mean should be reported to provide an indication of the fit of the continuous model simulation following the visual comparison of predicted and observed runoff hydrographs.

The PEV shows percentage differences in runoff volumes when compared to the observed runoff series. A value closer to zero indicates a better model fit. A negative value indicates that the model underpredicts the runoff volume, and a positive sign indicates the opposite. Criteria for assessing the catchment model performance were based on evaluating the model performance using NSE values against the criteria of 'good' models as developed by Moriasi et al. [27], based on review of different hydrological models. They evaluated the catchment models for flow predictions, with $NSE > 0.80$ as very good and with $0.70 < NSE \leq 0.80$, as good. Models with values $0.50 < NSE \leq 0.70$ were termed satisfactory, however a model with $NSE < 0.50$ was considered unsatisfactory for use in advanced studies.

### 2.7. Infiltration Systems Modeling

MUSIC was then employed to simulate the curbside leaky well systems using infiltration system node. The parameters for infiltration systems are based on geometry of the installed leaky well systems and as provided in Figure 2 of this document. The high bypass flowrate was based on laboratory trials (Figure 4) to understand the limitation of curbside leaky well systems to intercept the approach runoff based on corresponding flowrate. We constructed the full-scale road and curb model, similar to installed curbside leaky well systems at the NATA-accredited hydraulic laboratory (Australian Flow Management Group, University of South Australia) in University of South Australia. We measured the approach flowrate using the calibrated electromagnetic flow meter. using the services of a. The approach flows considered were 0.5 L/s, to 5 L/s, in increments of 0.5 L/s. The gradients considered were 0%, 0.5%, 0.8%, 1%, 1.2%, 1.5%, 2%, 2.5%, and 5%. We estimated the capture efficiency by measuring the time to fill a 20 L gradated bucket. The capture flow was measured three times for each slope/approach flow rate and the mean was reported (Figure 4).

### 2.8. Examining the Impacts of Aggregating Infiltration Systems on Stormwater Quality

We adopted a similar methodology for aggregating the infiltration system as reported by Elliott, Trowsdale, and Wadhwa [21]. However, we deviated from their approach and did not alter the travel times in links. Initially, we modeled each infiltration system individually and connected them to eight sub-catchments. The runoff deficit from the pre-installation model was noted. Then, we aggregated the infiltration systems in eight nodes and connected to eight sub-catchments, each aggregated node to each sub-catchment. This is somewhat tantamount to providing the street scale system instead of distributed storages. Then, we provided only one node, by aggregating 181 wells into one node, placed at the end of catchment, to intercept runoff from all the catchments. Figure 5 provides the summary of this process; we have used to select the aggregation model.

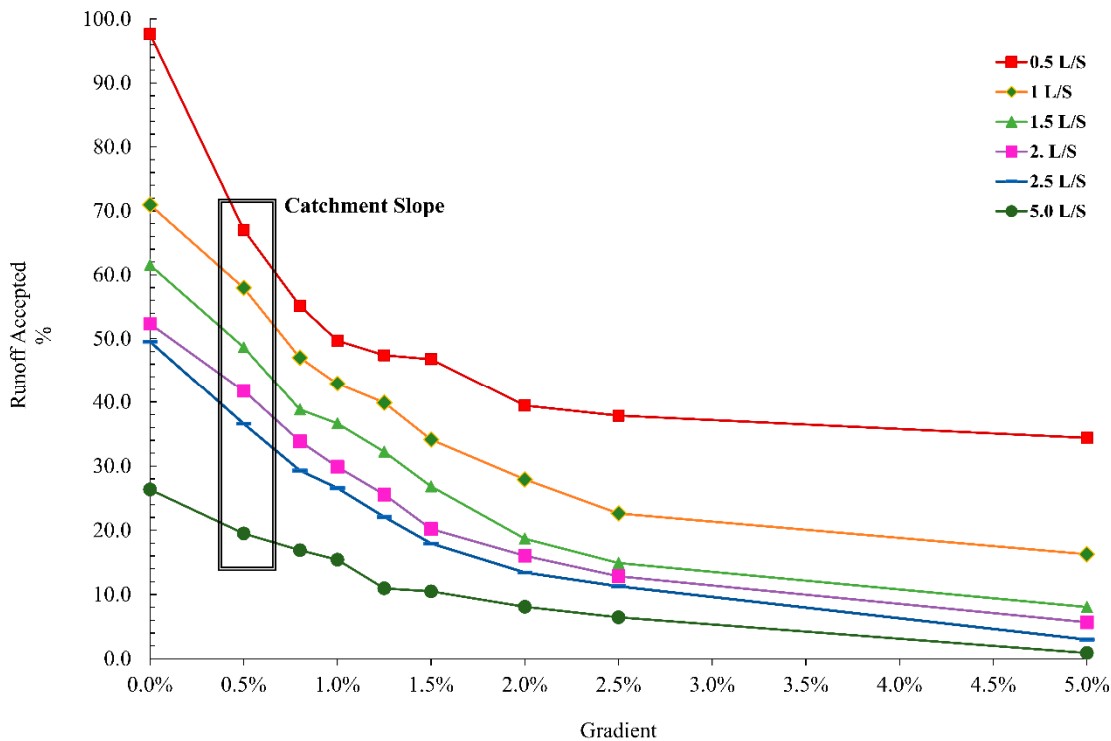

**Figure 4.** Analyzing the capture efficiency of TREENET inlet plate with approach flowrate. Acceptance rate for each flowrate corresponding to different gradients are presented in separate colors. The catchment gradient is highlighted in the figure.

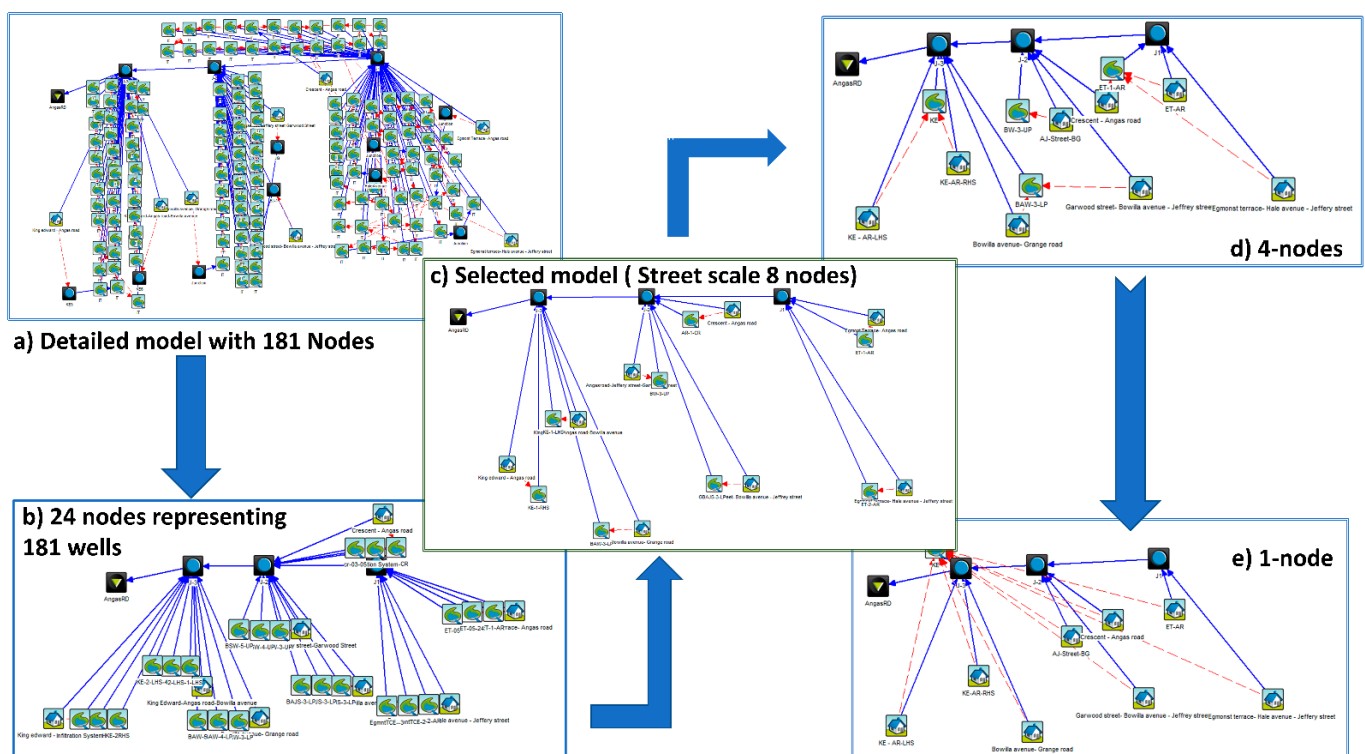

**Figure 5.** The graphical representations of aggregations methodology. The selected model placed in the center of the figure.(**a**) detailed model with 181 nodes (**b**) model with 24 nodes standing for 181 wells (**c**) street scale model with one node for each sub-catchment, this was selected for further analysis (**d**) model with 4 nodes standing for 181 well and (**e**) 1-node model.

We then carried out one-way ANOVA to test the presence of notable differences in means of these three configurations. The decision to adopt the aggregated levels was based on the hypothesis that if there is no significant difference between runoff from aggregated and distributed modeling effort, then the model with street level aggregation will be adopted for further studies. Table 2 summarizes the methodology adopted to convert each parameter into to represent aggregated nodes. The high flow bypass flowrate was based on prediction from PCSWMM model of the catchment for 4EY storms

**Table 2.** Methodology for estimating parameters for individual and aggregated values of the infiltration systems.

| Parameters | Detailed Model | Aggregation Methodology |
|---|---|---|
| No of infiltration wells for each sub-catchment | 181 nodes are modeled | Five levels of aggregation, based on combining wells in series in one street, street scale aggregation, combining all wells in one node |
| Inlet Properties—Low Flow By-pass (L/s) | Every low flow is assumed to be captured by the inlet | Aggregation should not impact the low flow bypass |
| Inlet Properties—High Flow By-pass (L/s) | Based on laboratory trials | Consider same for aggregated systems as well. Although this would change, however, as noticed the model was not sensitive to this parameter. |
| Storage and Infiltration Properties—Pond Surface Area (square meters) | Based on Excavated holes for the well | Sum of contributing nodes |
| Storage and Infiltration Properties—Extended Detention Depth (meters) | Based on length of inlet pipe | Inlet pipe remained same to reflect the similar capture efficiency of leaky wells. |
| Storage and Infiltration Properties—Filter Area (square meters) | Total area of the well based on excavated holes of 460 mm diameter | Sum of contributing nodes |
| Storage and Infiltration Properties—Unlined Filter Media Perimeter (meters) | Circumference of excavated hole | Sum of circumference of contributing nodes |
| Storage and Infiltration Properties—Depth of Infiltration Media (meters) | Maximum depth | Kept same as of individual systems to preserve the hydraulic head |

### 2.9. Water Quality Modeling

Using the calibrated and evaluated model, we then simulated the stormwater quality of the catchment to understand the catchment capacity to generate TSS, TN, and TP. MUSIC does not require a large set of parameters to simulate event mean concentrations of TSS, TN, and TP, as its default parameters are based on extensive review of local urban stormwater quality as reported in Duncan [19]. However, it must be mentioned here that most of these parameters are based on experimental trials from Brisbane and Melbourne. Land characteristics for different cities can impact the model outputs. Recently, Water Sensitive South Australia and Adelaide and Mount Lofty Ranges Natural Resources Management Board [20] have published separate set of guidelines (SA Guidelines for MUSIC Modeling) for selecting input parameters for MUSIC models, considering local climate and geology of South Australia.

MUSIC provides two options for estimating pollutant concentrations: The first option is to use the default parameters based on land use type and selected region, and stochastically generating pollutant loads based on specified probability distribution. The pollutant generation in MUSIC is based either on using mean concentration or log-normally generated distribution. In this study, using default parameters provided in MUSIC for stochastic generation of pollutants, we adopted the log normal distribution to stochastically generate the pollutants loads from the catchment models, as per recommendations of SA Guidelines for MUSIC modeling. These default parameters are based on research reported by Duncan [19]. Thus far. no study has reported the use of SA Guidelines for MUSIC

modeling on residential scale catchment models in South Australia to estimate the pollutant loading in stormwater. The second option is to estimate pollutant concentration based on user-specified pollutant concentration in connection with simulated runoff data. Pollutant generation is stochastic and based on mean and standard deviation of data entered by the user to represent local conditions.

The accuracy of model to represent the pollutant generation capacity was assessed by running *t*-tests on predicted and observed mean values and standard deviations. The simulated statistics were obtained from MUSIC outputs, by using flow-based sub-sample statistics. This option was selected based on the need to exclude no flow influence from the timeseries. The hypothesis was tested that there is no difference in mean concentration of a particular pollutant as observed from the catchments in comparison to against simulated MUSIC sample-based statistics.

### 2.10. Data Analysis

The study took the logical approach of first plotting the available observed sample results from the catchment along with international averages for these pollutants as reported by Duncan [19] and national averages as reported in Australian guidelines for water recycling [28]. The plots were developed for pre-installation and post-installation pollutant generation. The study also reviewed stormwater pollutants at the catchment outlet with respect to water quality criteria for governing environmental values, according to water quality criteria for receiving water bodies as developed by Environment Protection Authority, SA [29]. Note, however, that these guidelines in South Australia are only applicable to receiving water bodies as opposed to stormwater discharges from an urban catchment. The analysis of pollutants thus only provides an indication regarding the status of stormwater quality according to these ranges. Table 3 provides the summary of average pollutant concentrations as retrieved from mentioned documents.

**Table 3.** Range of stormwater pollutants as reported in international literature for stormwater quality. South Australia guidelines are also provided for receiving water bodies as reference.

| Pollutant | Range (mg/L) [19] | International Water Quality Mean (mg/L) [19] | National Water Quality Mean (mg/L) [28] | SA Guidelines for Receiving Waterbodies [29] |
|---|---|---|---|---|
| TSS | 50–500 | 180 | 99.73 | 20 |
| TN | 1.8–5.5 | 2.8 | 3.09 | 5 |
| TP | 0.08–0.8 | 0.24 | 0.480 | 0.5 |

### 3. Results

#### 3.1. Model Calibration

The MUSIC model was run using nine months of rainfall data. The model predicted a runoff volume of 6916 kL over this period, which was 14% more than the observed runoff for this period. The overall NSE value over the nine-month period was 0.53, which was deemed satisfactory to carry out further analysis according to criteria developed by Moriasi, Gitau, Pai, and Daggupati [27]. The model performed particularly well for some individual storms as extracted from the continuous timeseries, with NSE values ranging from 0.75 to 0.95 (Table 4 and Figure 6).

Based on the overall model criteria for individual storms extracted from the timeseries, the model was in a "very good" category. However, for some storms the model predicted a runoff volume more than 20%. Still during some storms, the model was not able to mimic the hydrograph satisfactorily, despite predicting accurate volume and peak. This is model limitation, which arose due to three possible causes: (1) it is hard to calibrate the continuous series, (2) limited routing options as offered by MUSIC, and (3) possible errors in field flow measures [21]. These limitations influenced the timing of hydrographs, due to which higher NSE value for continuous series was not achieved; however, overall, the model still performed satisfactorily.

**Table 4.** Goodness-of-fit statistics for individual storms as extracted from the continuous runoff series. The model performance during each storm was also evaluated based on NSE values.

| Event | NSE | PEP | PEV | Evaluation Based on NSE |
|---|---|---|---|---|
| 29 January 2016 | 0.89 | 4.96 | 3.77 | Very Good |
| 2 February 2016 | 0.93 | 5.17 | 0.54 | Very Good |
| 10 March 2016 | 0.88 | 13.35 | 15.86 | Very Good |
| 27 May 2016 | 0.94 | 4.13 | 8.70 | Very Good |
| 6 June 2016 | 0.90 | 1.37 | −10.12 | Very Good |
| 23 June 2016 | 0.87 | −14.83 | 10.25 | Very Good |
| 4 July 2016 | 0.75 | −7.54 | −23.13 | Good |
| 25 July 2016 | 0.95 | 10.90 | 8.29 | Very Good |
| 18 September 2016 | 0.95 | 15.78 | 8.37 | Very Good |

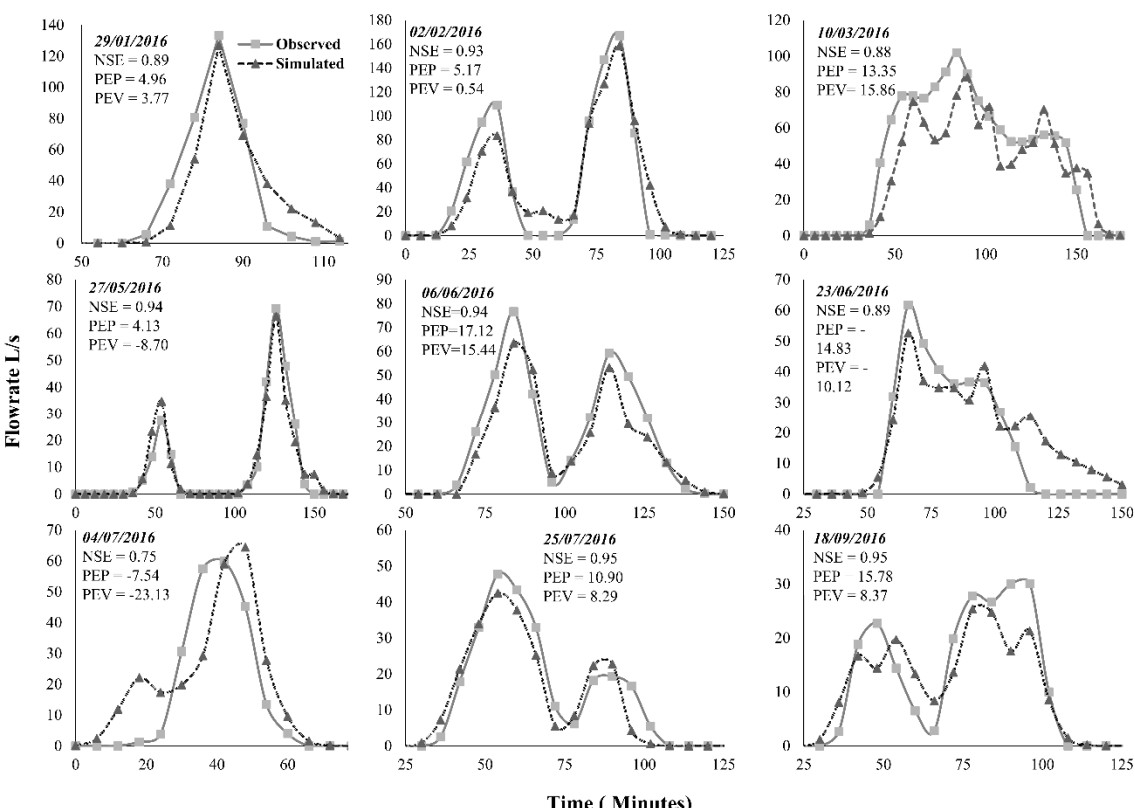

**Figure 6.** A comparison of the observed and simulated hydrographs of individual storms extracted from the continuous simulation runoff time series from the MUSIC model. Goodness-of-fit statistics are also presented with event data.

### 3.2. Water Quality Analysis

Water quality data from the monitored events is presented in Figure 7, with comparison to international [19] and Australian [28] mean stormwater quality. The results for TN concentration indicate that the catchment has produced an average of 2.81 mg/L across the five sampled events, which was less than the mean reported for Australia (3.09 mg/L). The figure also shows the monitored pollutants in comparison to SA Guidelines for receiving surface water quality [29]. We have seen that for TN and TP the catchment produced less pollutant concentration than that accepted by SA Guidelines to maintain the quality of receiving water bodies. However, for TSS, the load produced was approximately 10 times higher than the standards established by SA guidelines [29].

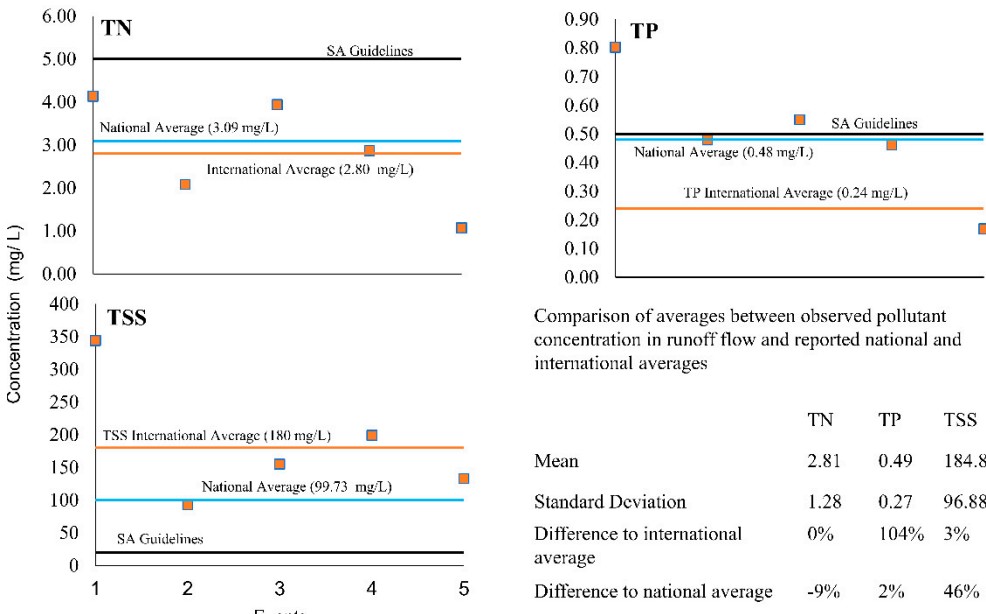

**Figure 7.** Characterization of stormwater quality of the case study catchment in comparison to national and international mean stormwater quality data.

### 3.3. MUSIC Pollutant Predictions

The MUSIC model performed adequately to simulate pollutant generated from the catchment based on comparing the output of the model with default stochastic parameters and monitoring data from the case study catchment. Figure 8 indicates that a *t*-test was not able to detect any significant differences between the distribution of monitored pollutants and the simulated model output. The largest variation between means was observed for TSS, however as indicated by p-value, this difference was not significant. Based on these results, we concluded that the model was suitably able to represent the stormwater quality of the case study catchment.

### 3.4. Leaky Well Aggregation

After evaluating the capability of the model to simulate water quality adequately, we used the catchment model and equipped it with infiltration systems. The catchment contains 181-distributed curbside leaky well systems, and to investigate how these systems could be simulated with less modeling effort by practitioners, we aimed to test how well the MUSIC tool could represent multiple devices with fewer model nodes by aggregating them. The parameters adopted to represent the aggregated well cases through several aggregation levels are presented in the Table 5. Aggregation of devices reduced the model runtime considerably, i.e., from approximately one hour to 5 min for generating outputs. This reduction in model runtime is useful and will become more critical if the objective is to calibrate the MUSIC model with distributed storages to the catchment outflows. An automated calibration tool, for example PEST, may undertook thousands of MUSIC simulations and corresponding evaluation before providing users with output parameters. Therefore, one hour of runtime will result in days of model simulation, which may not be a practical option.

The results of using these individual sets of scenarios are summarized in Table 6, and those of the *p*-value based on post hoc Tukey's multiple comparison test are shown in Figure 9. The results showed that with each level of aggregation there is an increase in the simulated performance of leaky well systems to retain runoff volume. The results of one-way ANOVA (F = 0.23, *p* = 0.91) indicated that this increase is not statistically significant. We further checked that if there is any significant difference between any groups, however, the figure indicates that these differences were not significant based on comparing the

original 181 well model to the aggerated model scenarios using multi comparison Tukey's test. Therefore, we concluded that MUSIC could withstand aggregation of large number of devices without variation in its performance.

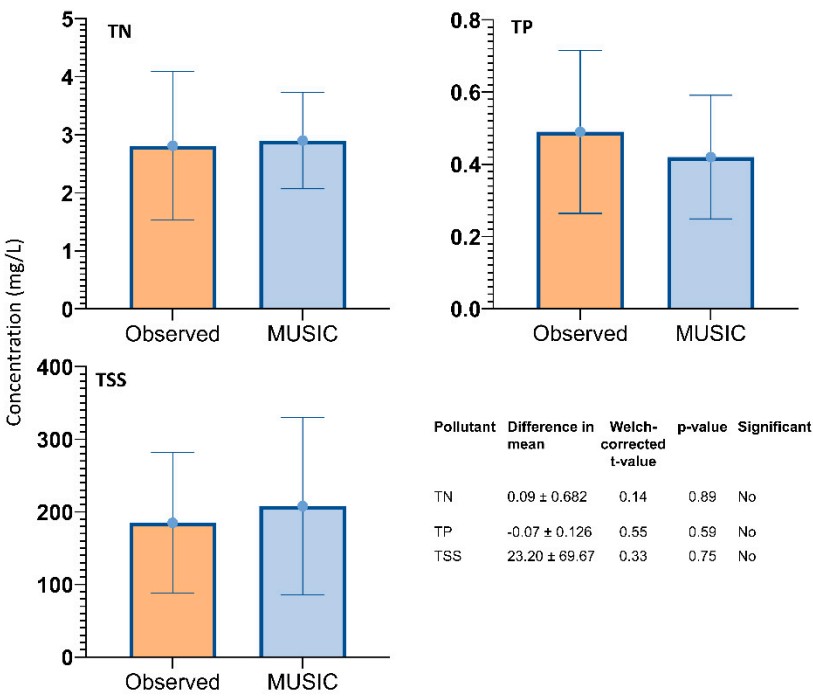

**Figure 8.** Analyzing the ability of MUSIC to simulate targeted pollutants using the stochastic generation process based on default values.

**Table 5.** Adopted values for different parameters due to aggregation of wells in combined nodes.

| Parameters | Wells Aggregated | | | |
|---|---|---|---|---|
| Contributing wells to each node | 7.54 | 22.63 | 45.25 | 181 |
| Inlet Properties—Low Flow By-pass (m$^3$/s) | 0 | 0 | 0 | 0 |
| Inlet Properties—High Flow By-pass (L/sec) | 5 | 5 | 5 | 5 |
| Storage and Infiltration Properties—Pond Surface Area (m$^2$) | 1.2 | 3.62 | 7.24 | 28.96 |
| Storage and Infiltration Properties—Extended Detention Depth (m) | 0.15 | 0.15 | 0.15 | 0.15 |
| Storage and Infiltration Properties—Filter Area (m$^2$) | 1.2 | 3.62 | 7.24 | 28.96 |
| Storage and Infiltration Properties—Unlined Filter Media Perimeter (m) | 10.86 | 32.54 | 65.16 | 260.64 |

We selected the model with eight nodes for further analysis. This decision was taken as there was only a 2% (Table 6) difference in catchment runoff volume when this model was compared to the catchment model with 181 individual wells. In all cases of aggregation, the total number of flow events remained 41 and none were nullified either due to presence of curbside leaky well system, or due to aggregation. Furthermore, by using the street scale model, i.e., one node for each subcatchment, we can extract more information regarding sensitivity of the performance of infiltration systems with regards to variations in catchment characteristics. This is because that in the street-scale model, by changing the catchment characteristics, in particular contributing impervious area, the effect of the change on the performance curbside leaky well systems can easily be established, whereas in case of lumped model, the change in one catchment characteristics may not manifest in noticeable change in the performance of curbside leaky well systems.

**Table 6.** The runoff volume produced with different levels of curbside leaky well aggregation.

| Nodes | Runoff Volume—Total Period (kL) | Difference in Comparison to Preinstallation Scenario | Mean Annual Runoff Volume (kL) | Standard Deviation |
|---|---|---|---|---|
| 181 | 6669.63 | 4% | 162.67 | 53.89 |
| 24 | 6535.56 | 6% | 159.40 | 53.38 |
| 8 | 6531.56 | 6% | 159.31 | 53.55 |
| 4 | 6410.40 | 7% | 156.35 | 53.17 |
| 1 | 6228.13 | 10% | 151.91 | 52.59 |

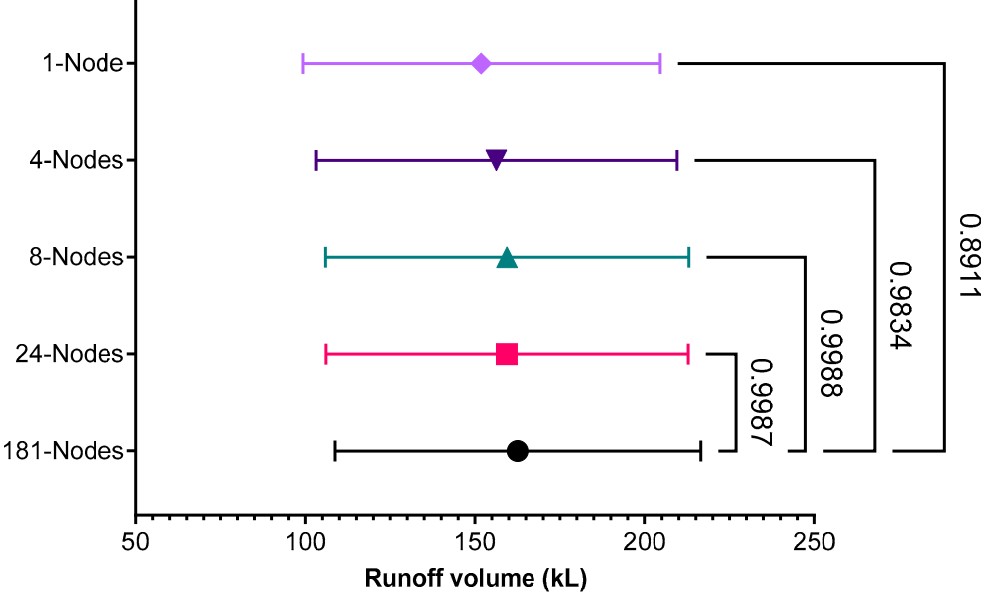

**Figure 9.** The effect of aggregation on the ability of the MUSIC tool to predict runoff volume from the case study catchment with 181 curbside leaky wells. The figure also shows *p*-values of post hoc Tukey's multi comparison test with reference to detailed model with 181 nodes to mentioned aggregation levels.

### 3.5. Performance of Leaky Wells to Reduce Pollutant Concentration in Catchment Outflows

The simulation results indicated that the presence of 181 leaky wells as installed in the case study catchment provided a marginal reduction in the annual pollutant discharges from the catchment. This is attributed to the limited storage capacity of installed wells (100 L) and the adoption of conservative exfiltration rates in clayey soils, resulting in frequent bypass of runoff from the wells. Pollutant removal of these devices is dependent on the runoff volume intercepted and infiltrated, as there is no other mechanism through which they can reduce pollutant loads. The average retention for the targeted pollutants was between 7.5% to 8.4% (Figure 10). While this does not appear to be a large retention, note that even this level of performance indicates that the curbside leaky wells intercepted 317 Kg of TSS over the simulation period of nine months. Which still contributed to lowering the sediment and nutrient transports to the receiving waterbodies, therefore protecting the marine environment by reducing algal growth.

### 3.6. Impact of Connected Impervious Area

Further investigation revealed that the performance of leaky wells to reduce pollutant loads is sensitive to the increasing contributing impervious area. The pollutant reduction showed strong correlation with impervious area, as shown in Figure 11. In this analysis, only the impervious area is considered, which is reasonable as WSUD devices are generally expected to intercept runoff from the impervious area. This analysis highlights the importance of storage capacity for the functioning of leaky wells in providing water quality benefits. With increasing connected impervious area, there is an increase in runoff bypass,

and as such the percentage of total runoff intercepted is reduced, and by association, so is pollutant load. For better performance of distributed storages, their service area needs to be estimated carefully so that a significant amount of runoff can be captured, which in theory will imply capture of larger amount sediments and associated pollutants.

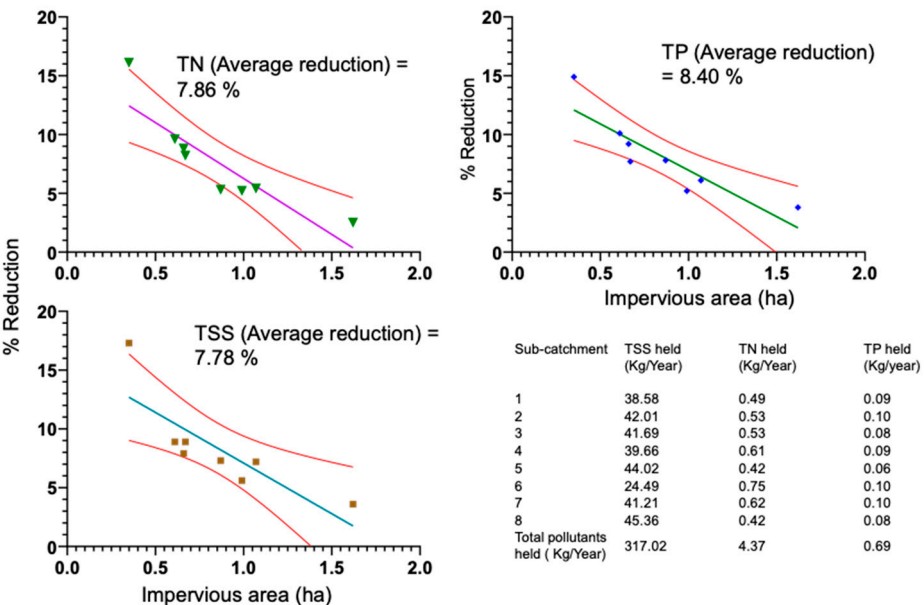

**Figure 10.** Performance of leaky wells to intercept pollutants from transporting downstream to receiving waters. The trend with increasing impervious are shown along with 95% confidence limits. The average reduction annually is also noted in the figure. The quantity of each pollutant held annually in the catchment, as per the MUSIC predictions are also shown.

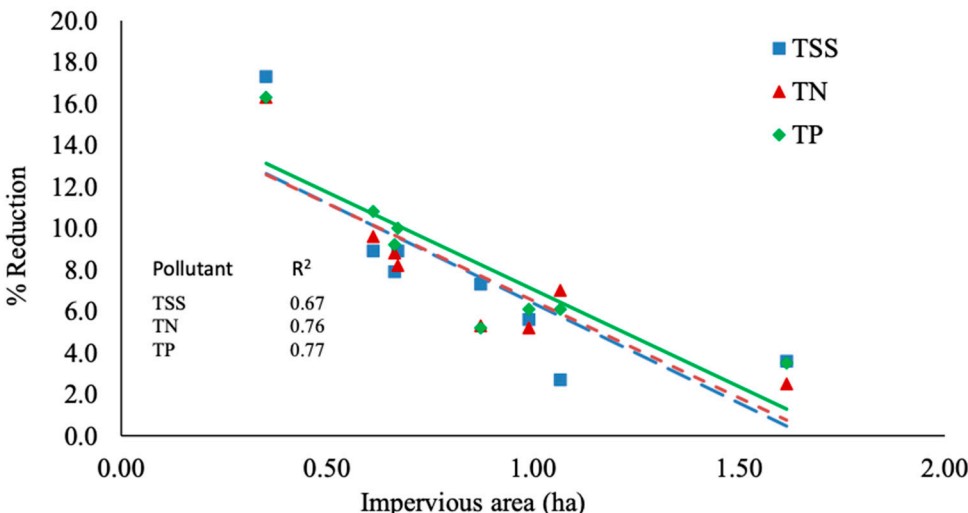

**Figure 11.** The relationship between pollutant reduction and contributing impervious area. The figure also indicates the correlation for TSS, TN, and TP with increasing contributing area. The pollutant reduction decreased with increased contributing impervious area.

## 4. Discussion

### 4.1. Stormwater Quality of Catchment Outflows

The generation of pollutants is influenced by different factors such as rainfall intensity, land type, and antecedent dry period. Their relevance is well established in previous studies, for example, Gustafson et al. [30]. Based on observed data, the average concentration of TN

in stormwater outflows from the catchment was 9% less than the national average, while TP was similar to reported national averages.

Interestingly, the mean average value of TN is even less than recommended TN concentration of nutrients as per SA guidelines for surface water bodies. These results agree with the conclusions of Lucke, Drapper, and Hornbuckle [4]. They also reported that levels of TN in urban stormwater from residential catchment may be less than previously considered. Though that study is based in Queensland, and thus its results should be corrected for traffic volumes, runoff productions, and air pollution, as per our monitored flows, this observation is apparently valid for case study catchment as well. However, one notable difference between the present study and one performed by Lucke, Drapper and Hornbuckle [4] is that the maximum area of seven monitored residential catchments is 7.46 ha. It could be the reason that their reported mean concentration of pollutants is significantly less (TSS = 54.4 mg/L, TN = 1.57 mg/L, & TP = 0.34 mg/L) than the observed data from this case study catchment. It is because, as we note from reported values of Lucke, Drapper and Hornbuckle [4], that their average catchment area is 3.46 ha. Therefore, per Hecate pollutants generation is TSS = 15.90 mg/L/ha, TN = 0.49 mg/L/ha, TP = 0.09 mg/L/ha. In comparison this catchment generated per hectare pollutant concentration in the following order: TSS = 10.53 mg/L/ha, TN = 0.16 mg/L/ha, TP = 0.02 mg/L/ha. Therefore, this full scale residential catchment has generated a smaller number of pollutants per hectare in comparison to the reported values of [4].

The results from field investigations did reveal that TSS was generally higher than national averages and guidelines for receiving water bodies. It could de due to the vegetated street footpaths which are abundant in the catchment. We based this inference on the analysis of Figure 11, where it can be seen that of three pollutants, TSS showed least correlations with increasing impervious area. Therefore, increased number of TSS could be due to the erosions of soils from footpath vegetations. The unvegetated pervious area can also contribute large number of sediments to the runoff flow resulting as due to wear, caused by the runoff flow. The reason could be the existence of large lawn areas in the catchment. The literature has identified lawns as areas with high potential to release nutrients and sediments to runoff flows [31]. The contribution however will only occur if the lawns do produce runoff despite their higher ability to retain runoff [32]. The decompositions of grass and other street vegetation is another possible cause of increased nutrient concentrations [31]. Furthermore, use of fertilizers would add more nutrients to the discharge from garden areas. In this regard the interaction of green areas with grey areas is also of concern, with potential release of organic matter, which facilitate the transport of pollutants to the drainage flows [32]. Similarly, maintenance related construction activities can mobilize high volumes of sediment. Still, another source of potential mobilization of sediment could be the movement of automobiles [33]. These reasons could have resulted in higher concentrations of some pollutants than other studies of this nature.

*4.2. Performance of MUSIC*

The results verified the ability of MUSIC to represent the catchment runoff peak flow and volume in a satisfactory manner with appropriate parameters. However, note that MUSIC could only simulate the continuous time series to meet minimum requirements of good model (NSE > 0.50). One of the reasons for this could be the fact that with simplistic impervious losses, as are currently in MUSIC [25], may have led the inconsistent model behavior for small events, resulting in overprediction of several small runoff events on the constant basis. This in combination with the limited hydraulic routing options resulted in model not entirely mimicking the timing of the hydrograph as is evident in event of 04 July 2016 as shown in Figure. Furthermore, as Figure 6 illustrates that the model was consistently out of synchronization in lower limb of the hydrographs (see events 23 June 2016 and 25 July 2016. This resulted in compounding error and manifested with lower NSE value for the simulated flow series. Ignoring some large storm events of magnitude 0.2 exceedances per year also contributed to lowering overall NSE of the flow series.

MUSIC also performed satisfactorily to generate flow weighted pollutants using the built-in stochastic pollutant generations tools. The distribution of simulated pollutants did not vary significantly from the pollutant loads observed from the case study catchment. It is because the water quality predictions are dependent on the estimation of runoff volume as results from Section 3.1 demonstrated, MUSIC predicted the runoff volume accurately.

Another study finding is that aggregation of infiltration devices did not greatly affect the model performance. With increased numbers of devices represented in a single node of infiltration, there was a noticeable improvement in runoff retention, but we considered the increase statistically insignificant, based on the results of one-way ANOVA and pos hoc Tukey's multi comparison test. The results of aggregation however can be different for street scale systems for example raingardens as reported in Myers et al. [34]. Such systems may entail different challenges. The finding can still prove valuable for stormwater designers who may wish to ascertain the effectiveness of implementing distributed infiltration systems with different storage volumes across much larger catchment areas. Aggregation of leaky wells provides a way to simulate the performance of systems in an efficient and effective way without losing essential accuracy of the detailed model.

### 4.3. Performance of Distributed Curbside Leaky Well Systems

The limited combined storage capacity of 18 kL of curbside leaky well system proves to be insufficient to hold significant amount of runoff volume to result in significant reduction in pollutant outflows from the catchment. The performance of curbside leaky well systems is needed to be evaluated under load reduction targets for different states. The pollutant reduction targets as recommended for different WSUD, in South Australia are to reduce TSS by 80%, TP by 60%, and 45% reduction in annual TN loads [35]. The installed leaky well systems with their limited storages were not able to meet pollutant load reduction targets. The actual performance could have been better if the influence of first flush can be considered in their performance. At present, however, we do not have the data to include this aspect in this study. The performance of the curbside leaky well systems can potentially be improved by installing leaky wells either larger in storage volume or by increasing their per hectare density. Both of these options will reduce the amount of contributing area to storage ratio. The importance of this ratio and contributing impervious area, in the performance of distributed leaky well systems, was reported in Section 4.3. The results showed strong correlation between declining performance of leaky wells with increasing contributing impervious area. Reducing the ratio of contributing impervious area to storage rations can potentially improve the performance of distributed storages, and it is only possible by installing systems of large storage volume. However, it must also be considered that their primary objective is to provide passive irrigation to street trees during periods of drought. The stormwater quality improvement is in addition to this, which make these reductions as added advantage.

Another reason could be that as major source of nitrates is from the street vegetation, the leaky well with small inlet orifice may not be able to capture large pollutant sizes. No data exist which has evaluated the inlet design to estimate the limits of captured sediments size. However, clogging can potentially become the associated problem which can occur if these systems are designed to capture sediments with large particle size. Despite not meeting the target, curbside leaky well systems were able to reduce 317 kg of TSS over nine months of simulating period, which by considering that stormwater improvement is only added advantage, makes this statistic acceptable.

### 5. Conclusions

The study has evaluated the catchment outflows from a residential case study catchment and investigated the potential of curbside leaky wells to reduce runoff volume and pollutant loads using the MUSIC software tool. The study found that MUSIC performed satisfactorily to produce the observed catchment runoff volume and peak flow rates for the 17-ha case study catchment, but the model can benefit from enhanced hydraulic features.

MUSIC also performed adequately to simulate water quality using the in-built stochastic pollutant generation algorithm. Furthermore, the MUSIC software adequately represented the impact of distributed curbside leaky well storages and allowed for aggregation of large number of devices with minor impacts on model performance. The pollutant load can be influenced by the contributing impervious area and therefore, this study showed relatively higher pollutant loads in comparison reported pollutants in national studies.

The performance of 181 leaky wells distributed over the 17-ha1 case study catchment was only able to reduce up to 8% of pollutant load per year. Based on this result, the study concluded that for meaningful pollutant reductions, distributed storages of larger volume are required. We also recommend that, considering the impact of stormwater on Adelaide coastal waters, the established catchments in Adelaide should set up instrumentations to monitor the stormwater quality and estimated output of pollutants. In terms of leaky well inlets, we also recommend that the captured stormwater should be sampled and tested for particle size and pollutant concertation. This will inform the system performance in capturing unit mass of pollutants and will also provide information regarding their potential in improving stormwater quality.

**Author Contributions:** H.S. and B.M. were involved in preparing the study design and execution. H.S. prepared the draft and carried out all the analysis under the supervision of B.M. and G.H. T.J. provided access to the study site and planning of instrumentation for field investigations. J.B. assisted in carrying out the statistical analysis as are reported in this paper. H.M. assisted in preparing the figures in required format and preparing the draft. All authors have read and agreed to the published version of the manuscript.

**Funding:** This research received no external funding. The City of Mitcham provided the monitoring instrumentation and stormwater quality data.

**Data Availability Statement:** Water quality results are available from https://greenadelaide.waterdata.com.au/WaterQuality.aspx?sno=W5040082&Report=trConcentrations (accessed on: 6 November 2021). Also, runoff flow and rainfall series in daily formats can be downloaded from this link. The require runoff data and MUSIC model is available upon request from the corresponding author for all reasonable requests.

**Acknowledgments:** The authors appreciate the administrative support provided by University of South Australia to provide corresponding author with logistics to make possible this research. The authors are also thankful to City of Mitcham and NRM board for installing monitoring equipment in the catchment, which make possible the collection of field data. In addition, we are also thankful to Jason Homa of Australian Flow and Management Group (AFMG) for arranging hydraulic testing of curbside inlets.

**Conflicts of Interest:** The authors declare no conflict of interest.

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
