# Peer review of "Characterizing the Stormwater Runoff Quality and Evaluating the Performance of Curbside Infiltration Systems to Improve Stormwater Quality of an Urban Catchment"

_water, doi:10.3390/w14010014_

Round 1
Reviewer 1 Report
The manuscript deals with the problem of the conveyance of urban pollutants to receiving water bodies and reports on the efficacy assessment of distributed curbside leaky well systems to improve the water quality. A real case study is investigated, i.e., a typical urban catchment in Australia in Hawthorn, an inner suburb of Adelaide, South Australia.
The article is an original contribution and the analysis of a real case study strengthens the research. The topic is of interest for the readership of the Water journal.
English language is clear, the presentation is good; anyway, I have detected some criticisms in the text that should be properly addressed.
The Authors can benefit from the comments below to improve their paper. These have to be accomplished before manuscript acceptance.
Title
Title is appropriate but not concise; it could be shortened.
Abstract
The abstract is concise and reflects the content of the article. It includes the main findings of the study.
Introduction
Aims of the study are properly clarified in the Introduction. Relevant references are included.
Lines 50-52: Concerning the theme of reducing discharge of stormwater runoff of pollutants from urban catchment to receiving waters, the Authors are recommended to include, among others, the following reference as part of the introductory discussion:
- Todeschini S., Papiri S., Ciaponi C. (2018). Placement strategies and cumulative effects of wet-weather control practices for intermunicipal sewerage systems. Water Resources Management, 32(8), 2885-2900, DOI: 1007/s11269-018-1964-y.
This study investigates the rainfall-runoff process and the pollutant dynamics on theoretical residential catchments and intermunicipal sewer systems in order to characterize the hydraulic and environmental performance of possible types of sewerage system and to compare the effectiveness of different wet-weather control schemes for intermunicipal sewerage systems. A comprehensive investigation on the placement strategies and the cumulative effects of wet-weather control practices is carried out over a broad watershed as a key preliminary step in addressing the safeguard of receiving waters in developing and urban areas.
Line 83: I suggest adding a reference for the Model for Urban Stormwater Improvement Conceptualization (MUSIC).
Materials and Methods
This section is clear and adequately detailed. The provided figures and tables are clear and necessary for the presentation.
Line 134: The reference to Figure 1 is lacking in the text.
Lines 170-175: the maximum diameter value seems rather small considering the total contributing area of 17.45 ha. What is the return period adopted for the stormwater drainage pipes?
Line 181: I suggest adding “area-velocity” before “flow meter”
Lines 231-233: Is there some study that compares MUSIC and EPA SWMM modelling results? This would be of interest.
Line 249: Please, specify the different model runtime with 8 nodes as opposed to 108 nodes.
Line 260: Replace “were” with “was”.
Lines 269-293: It is not clear the adopted objective function. The fitness of the model to observed data is in terms of total runoff volume of the representative flow events? Please, clarify.
Lines 338-339: Please, complete the final sentence on the fact that statistical tests are based on mean.
Lines 352-353: In Table 2 or in the caption of Table 2 the reference for the provided ranges should be included.
Lines 367-369: Why an electromagnetic flow meter is adopted in measuring the approach flow rate? Electromagnetic flow meter is not suitable for measuring surface-free flow.
Results
This section is clear and presented in a logical sequence. The provided tables and figures are clear and useful for the presentation of the results.
Line 436: In Figure 7 I suggest including a legend valid for the three graphs for the colors of the guideline threshold and national/international average.
Line 478: It should be useful to specify (in the Materials and Methods Section) how a rainfall event is selected (i.e. minimum total depth, inter event time between contiguous rainfall events).
Lines 512-518: The storage capacity should be fixed based on the contributing impervious area.
Discussion
This section is interesting and useful. Discussion is supported by relevant and pertinent references.
Line 536: Remove “,” between “The” and “decompositions”. Typo error.
Lines 583-584: “The actual performance could have been better if the influence of first flush can be considered in their performance”. I agree with this comment. The detailed analysis of the pollutant dynamics in wet-weather runoff is crucial for a more accurate performance assessment of the curbside leaky well system.
Conclusions
Conclusions seem reasonable and are supported by the results.
References
One reference is suggested in Introduction Section on the approaches for the reduction of pollutants discharged in wet-weather from urban catchments to receiving waters. Apart from this reference, based on my knowledge, no important reference is missing.
Author Response
Thank you for your comments. We have now incorporated the majority of your comments in the revised manuscript. Please see the attached files

Reviewer 2 Report
The article is interesting but needs to be organised to increase its readability. Noteworthy is the fact that the authors use several tools: statistical, modelling as well as results from field measurements. In my opinion, the latest ones are the most important, because they were the basis for the next steps, including model calibration and verification. For these reasons, in my opinion, the results of field research should be more emphasized in the article. The authors conduct a 9-month monitoring and consequently show only a few selected results of the research in the article. There is a feeling of insufficiency. The issues of the quality of runoff from the measurements should be more elaborated in the article.
Comments related to the content of the article are the following:
- Abstract: Some of the sentences require modification. The term receiving water bodies is used three times. Please try to find a replacement for this expression;
- Please state in the text whether the MUSIC program used is an open source type of software or commercial one;
- Chapter 1.1 Research objectives should be connected with the Introduction;
- For the purpose of the article, the authors write that they compare the results to national averages. In a later part of the article the reader reads that also to international averages. The content must be unified;
- In chapter 2.1. I propose to replace the paragraphs. As first give " Field investigations and monitoring were conducted in a 17.45 ha urbanized...", as second "Adelaide has cooler temperate waters in its coastlines, which offer a conducive.."
- The methodology needs to be organised. First, all issues concerning the quantity of runoff should be described, then the quality of the runoff, and then the modelling in the same order: first quantity, then quality;
- Line 237: Specify source of world-based data of different land types;
- The authors quote MUSIC guidelines once as SA MUSIC guidelines and once as SA guidelines. Please unify;
- Lines 299-300: the authors state that most of the parameters are based on experimental trials from Brisbane and Melbourne. The question is why were data not used for the case study (Adelaide) for which monitoring was conducted?;
- Table 2 should be more detailed with more literature and data from it.
- Table 2. There is SA guidelines [29] and should be SA guidelines [18];
- Figure 5 is not readable;
- Figure 8: Each graph should indicate the quality parameter to which it relates;
- The analysis of the results is not based on the values obtained from the surveys. These values are only included in the tables and graphs, but lack detailed interpretation in the text;
- The situation as above is also in Conclusions. These need to be supplemented with values from the analyses.
Author Response
Hi We have now incorporated the majority of the comments in the revised manuscript

Round 2
Reviewer 1 Report
The manuscript has been significantly improved following the recommendations of the Reviewers; all my concerns have been addressed and convincingly justified.
Reviewer 2 Report
Authors have made efforts to make the article more informative for the Reader and better in terms of content. In my opinion it could be considered for publication in the Water journal.